# Endoscopic Imaging for the Diagnosis of Neoplastic and Pre-Neoplastic Conditions of the Stomach

**DOI:** 10.3390/cancers15092445

**Published:** 2023-04-25

**Authors:** Bruno Costa Martins, Renata Nobre Moura, Angelo So Taa Kum, Carolina Ogawa Matsubayashi, Sergio Barbosa Marques, Adriana Vaz Safatle-Ribeiro

**Affiliations:** 1Endoscopy Unit, Instituto do Cancer do Estado de São Paulo, University of São Paulo, São Paulo 01246-000, Brazil; 2Fleury Medicina e Saude, São Paulo 01333-010, Brazil; 3Endoscopy Unit, Hospital das Clínicas da Faculdade de Medicina da Universidade de São Paulo, University of São Paulo, São Paulo 05403-010, Brazil

**Keywords:** gastric cancer, endoscopy, artificial intelligence, early detection of cancer, early diagnosis

## Abstract

**Simple Summary:**

Gastric cancer has a poor prognosis when diagnosed in advanced stages, but curative treatment is possible if an early diagnosis is made. Endoscopy represents an essential tool for the detection of early neoplastic and pre-neoplastic gastric lesions and for surveillance. Many endoscopy imaging technologies have been developed to increase diagnostic accuracy. In this review, we summarize these endoscopy technologies.

**Abstract:**

Gastric cancer is an aggressive disease with low long-term survival rates. An early diagnosis is essential to offer a better prognosis and curative treatment. Upper gastrointestinal endoscopy is the main tool for the screening and diagnosis of patients with gastric pre-neoplastic conditions and early lesions. Image-enhanced techniques such as conventional chromoendoscopy, virtual chromoendoscopy, magnifying imaging, and artificial intelligence improve the diagnosis and the characterization of early neoplastic lesions. In this review, we provide a summary of the currently available recommendations for the screening, surveillance, and diagnosis of gastric cancer, focusing on novel endoscopy imaging technologies.

## 1. Introduction

Gastric cancer (GC) is the fourth most common cause of cancer deaths worldwide, with 768,793 cases and an estimated incidence of 10,839,103 new cases yearly, according to The International Agency for Research on Cancer. Eastern Asia has the highest incidence, accounting for 60% of the cases [1]. According to gender, both the incidence and death rate in males were more than twice that in females. Most GC cases occur in patients older than 45 years of age, with a higher incidence at 65 to 70 years.

Although advanced GC is associated with a poor prognosis and high mortality, early detection and treatment have a good prognosis, with 5-year survival rates higher than 95% [2]. Unfortunately, more than 70% of GCs in Western countries are diagnosed in advanced stages [3].

Histologically, GC is classified as diffuse (composed of non-cohesive cells) or intestinal types (gland-forming), with different epidemiological patterns [4]. *Helicobacter pylori (H. pylori*) is considered a risk factor for non-cardia GC, with nearly 89% of cancers associated with this pathogen infection [5], mainly intestinal-type cancer. According to Pelayo Correa’s model of carcinogenesis, a cascade of events beginning with active chronic inflammation may progress to multifocal atrophic gastritis (AG), intestinal metaplasia (IM), dysplasia, and carcinoma [6]. The eradication of *H. pylori* infection has been associated with a significant reduction in the incidence and mortality of GC, especially among patients younger than 50 years of age [7]. In a meta-analysis, a pooled incidence rate ratio of 0.53 (95% CI: 0.44−0.64) was observed comparing individuals who received *H. pylori* eradication with individuals who did not receive eradication therapy [8].

Esophagogastroduodenoscopy (EGD) is the gold-standard exam for the diagnosis of neoplastic and pre-neoplastic gastric conditions, such as AG and IM. However, in a meta-analysis of 22 studies, the authors demonstrated that nearly 10% of GCs were potentially missed during white-light endoscopy (WLE), mainly adenocarcinomas located at the gastric body. Predictive factors for diagnostic failure were a younger age (<55 years), female, advanced atrophy, adenoma, ulcer lesions, and an insufficient number of biopsies [9].

*Image-Enhanced Endoscopy* (IEE) has been used to overcome the diagnostic limitations of standard endoscopy. IEE refers to various methods, such as dye chromoscopy, high-resolution imaging, virtual chromoscopy, and artificial intelligence. IEE provides a better assessment of the mucosal surface, increasing the detection of subtle changes and improving the diagnosis of pre-neoplastic lesions [10,11,12]. In this article, we will discuss the technical measures and imaging technologies that can be adopted to increase endoscopy diagnostic yield.

## 2. Indications for Endoscopic Screening of Gastric Cancer

In Eastern countries, radiographic screening programs for GC diagnosis were implemented in the 1960s, reducing its mortality [13]. Nowadays, screening methods include radiography and EGD. A cohort study comparing the two methods showed that subjects screened by EGD had a 67% reduction in GC mortality compared with subjects screened by radiography (RR 0.327; 95% CI, 0.118–0.908) [14]. In South Korea, GC screening by EGD every 2 years was shown to be associated with a significant (≥80%) reduction in GC mortality, while for those undergoing radiographic examinations every 2 years, the reduction in mortality was only 20% [7]. In a metanalysis, EGD screening detection of GC (0.55%, 95% CI 0.39–0.75%) and early-GC (EGC) (0.48%, 95% CI 0.34–0.65%) was superior to radiography screening (GC 0.19%, 95% CI 0.10–0.31%; EGC 0.08%, 95% CI 0.04–0.13%) [15].

In Western countries, where the incidence of GC is lower than in Eastern countries, screening focuses on high-risk patients with AG or IM. The management of epithelial precancerous conditions and lesions in the stomach (MAPS) guideline recommends the use of a staging system in patients with AG and/or IM, such as the Operative Link on Gastritis Assessment (OLGA) or the Operative Link on Gastritis Assessment based on Intestinal Metaplasia (OLGIM) systems [16,17]. Patients with extensive atrophy and/or IM, i.e., affecting both antral and corpus mucosa, should be identified and sampled as they are considered to be at higher risk for GC [17]. These patients should be followed-up with a high-quality EGD every 3 years. In patients with a family history of GC, close follow-up is suggested (e.g., every 1–2 years after diagnosis). The association of high-risk OLGA stages (III/IV) with GC was demonstrated in a meta-analysis including 6 case–control studies and 2 cohort studies (2.64 and 27.7 times higher chance of GC than lower OLGA stages, respectively). Among patients with OLGIM III/IV, the risk of GC was 3.99 times higher (95% CI 3.05–5.21) [18].

The American Gastroenterological Association (AGA) suggested EGD surveillance every 3–5 years in patients at high risk for GC, including those with advanced AG, extensive and incomplete IM, a family history of GC, and new immigrants from areas of high risk, such as East Asia or South America. The AGA also recommended testing and the eradication of *H. pylori* infection among individuals with IM [19,20]. Similarly to the AGA, the American Society of Gastrointestinal Endoscopy (ASGE) recommended surveillance for high-risk individuals with IM, as well as testing and treating *H. pylori* infection [21].

The British Society of Gastroenterology emphasized the importance of IEE for the diagnosis of pre-neoplastic lesions. The society advised a baseline endoscopy among individuals with laboratory evidence of pernicious anemia, such as vitamin B12 deficiency, and positive gastric parietal cell or intrinsic factor antibodies [22].

## 3. White-Light Endoscopy (WLE)

EGD has the possibility of identifying pre-neoplastic mucosal changes, detecting early-stage GC, and reducing cancer-related mortality by diagnosing and treating gastric mucosal alterations and/or EGC. Under WLE, EGC should be suspected in the presence of mucosal surface irregularity and/or mucosal coloration changes. Spontaneous bleeding, pallor coloration, and alterations in the mucosa surface and in light reflection should raise a concern about neoplastic lesions, as shown in Figure 1.

Early gastric cancer is defined as a lesion confined to the mucosa and submucosa (T1), regardless of lymph node involvement. These lesions usually manifest as superficial lesions (type 0), which can be subclassified into polypoid (Type 0-I), flat (Type 0-II), and excavated (Type 0-III). Flat lesions are categorized as slightly elevated (0-IIa), completely flat (0-IIb), or slightly depressed (0-IIc). Superficial tumors with two or more components should have all the components described (e.g., 0-IIa + IIc) [23] (Figure 2).

A high-quality level of endoscopic examination is imperative to make a proper diagnosis of early neoplastic lesions. The use of pre-endoscopy medications (mucolytic and defoaming agents), high-definition endoscopes, adequate inspection time, obtaining index images, and the application of the MAPS biopsy protocol when AG and chronic inflammation are suspected have been recommended by the European Society of Gastrointestinal Endoscopy (ESGE) guidelines [24,25].

### 3.1. Mucolytic and Defoaming Agents

To achieve a proper mucosal evaluation during EGD, the cleansing of mucus, bubbles, and foam is important. The most common mucolytic agents used worldwide are Pronase^®^ and n-acetylcysteine, and both are used to eliminate gastric mucus using non-osmotic solutions. Simethicone (activated dimethicone) is a commonly used defoaming agent that decreases the surface tension of gas bubbles without significant adverse interactions. It can improve the mucosal observation when used 20 min before the procedure [26].

### 3.2. Antispasmodic Agents

Antispasmodic agents, such as cimetropium bromide, scopolamine, and hyoscine N-butyl bromide, can reduce peristalsis and may be used during EGD [27]. However, there is no scientific evidence supporting the benefits of antispasmodic agents in the detection rate of upper gastrointestinal (GI) neoplasia. In addition, patients may have adverse reactions to these drugs, such as arrhythmia, benign prostate hypertrophy, and glaucoma. Thus, its use should be selective and at the discretion of the endoscopist.

### 3.3. Inspection Time

Limited consensus exists about the optimum inspection time for EGD compared to the colonoscopy quality protocol recommendation (withdrawal time of 6 min or more). In a retrospective Korean study [28], endoscopists who dedicated at least 3 min to evaluate the gastric mucosa detected more gastric adenomas or cancers than faster endoscopists (0.28% versus 0.20%, respectively; *p* < 0.01). A Japanese study showed that faster endoscopists, with a mean inspection time below 5 min, may overlook neoplastic lesions in the upper gastrointestinal tract. In this study, endoscopists were classified into fast (<5 min examination), moderate (between 5–7 min), and slow (>7 min) groups. The odds ratio of diagnosing neoplastic lesions was 1.90 for the moderate group and 1.89 for the slow group compared to the fast group (*p* = 0.03 and *p* = 0.06, respectively) [29].

### 3.4. High-Resolution Endoscopes

The resolution of an image depends on the pixel density, which is directly associated with the capacity to distinguish two adjacent points. Higher pixel density endoscopes provide higher imaging resolution. Standard endoscopes produce a signal image with a resolution of 100,000 to 400,000 pixels. High-resolution or high-definition endoscopes generate images with up to 1,000,000 pixels [30]. The development of high-resolution endoscopes enabled to distinguish subtle mucosal surface details compared to standard endoscopes, allowing more accurate suspicious diagnoses and targeted biopsies [31].

### 3.5. Obtain Index Images

The requirements of minimum photo documentation vary from each endoscopy society recommendation. While USA guidelines do not specify the minimum number of EGD images to report, the ESGE suggested at least ten images indicating the anatomical index [32]. In Japan, a systematic screening protocol for the stomach included 22 images, with the rationale to study the full gastric mucosa and avoid blind spots [33]. The World Endoscopy Organization proposed a total of 28 image areas, including the hypopharynx [34]. Adherence to a standardized photo protocol may increase the EGD neoplastic detection rate because the endoscopist must examine the entire stomach, including potential blind spots.

### 3.6. Target Biopsies of Suspicious Lesions

Target biopsies improve the diagnosis of GC. The ESGE recommends at least six biopsies of suspected advanced GC [35]. For EGC, only one to two targeted biopsies are recommended to avoid scars or submucosal fibrosis, as it is best staged and treated by endoscopic resection [36].

Moreover, some conditions, including chronic AG or IM, carry a higher risk for GC. The updated Sydney system recommended at least five biopsies: two from the antrum (greater and lesser curvature, 3 cm from the pylorus), one from the incisura, and two from the gastric body (from lesser curvature and greater curvature) [37]. The MAPS I and II guidelines question the necessity of sampling the incisura, as it yielded minimal additional diagnostic information, with more costs [17]. The inclusion of an incisura biopsy, however, increased the proportion of patients classified as high-risk stages (OLGA III/IV or OLGIM III/IV) [38,39]. It is also important to label the biopsies from different sites, and to apply a validated staging system, such as OLGA or OLGIM.

## 4. Chromoendoscopy

Chromoendoscopy (CE) is an IEE modality consisting of spraying dyes on the mucosal surface to improve visualization of the lesions under investigation. The use of CE in the screening of malignant and premalignant lesions may increase the detection rate and provide a better understanding of the lesion boundaries and microsurface, which helps differentiate benign or inflammatory from suspected malignant conditions and to determine the adequate region for biopsy. CE has a relatively low cost and it can be used in any endoscopy unit. CE with acetic acid, methylene blue, and indigo carmine are the main dye spray techniques to improve the diagnosis of GC. CE had higher accuracy compared to SD-WLE for the diagnosis of EGC (*p* = 0.005) and premalignant lesions (*p* = 0.001) [10].

### 4.1. Acetic Acid

The acetic acid solution is used in a concentration of 1.5 to 3%. After spraying it into the gastric mucosa, an acetowhite reaction is seen immediately: the structure of the cellular protein is reversibly altered and the superficial pH is lowered in the mucosa, causing a white reflection. The intensity of the whitening differs for normal mucosa and IM, as well as the duration for cancerous and non-cancerous tissue, disappearing earlier in the carcinoma. This creates a contrast between the regular mucosa versus IM and the pinkish cancer lesion surrounded by non-neoplastic tissue [40,41], as seen in Figure 3.

### 4.2. Methylene Blue

Ida et al. first described the methylene blue staining in endoscopy to improve the diagnosis of EGC. Methylene blue is absorbed by the small bowel and colon mucosa and is not absorbed by the stratified squamous epithelium of the esophagus or normal gastric mucosa [42]. The effectiveness of methylene blue, used as a 5% solution, is to highlight subtle mucosal changes through the staining of IM within the stomach. This vital staining dye improves the accurate delineation of the anatomical extent of histological abnormalities in the stomach. This strategy improves the accuracy of mapping IM and guides target biopsies.

### 4.3. Indigo Carmine

CE with indigo carmine dye was first described by Tada et al. [43] in 1976. This contrast staining dye is not absorbed by the mucosa. It is used to enhance crevices and valleys, defining irregularities in the mucosal architecture more accurately (Figure 4). Indigo carmine is used in a concentration of 0.2–0.5% and is an important modality to classify and delimit gastric lesions (e.g., Figure 4).

## 5. Virtual Chromoendoscopy

Virtual or electronic chromoendoscopy are image-processing techniques capable of enhancing mucosal surface patterns. It is easy to use because it is activated by pressing a button in the endoscope, being less time-consuming than conventional dyes. Currently, there are several types of electronic IEE, which can be basically separated into two methods: post-processed images (ex., FICE and i-SCAN), which electronically select filters and reconstruct images, and pre-processed images, which use optical filters (ex., NBI, LCI, and BLI).

The basic principle consists of different tissue wavelengths absorption according to the depth of penetration. Narrow-band imaging (NBI) developed by Olympus (Olympus Medical Systems Co., Ltd., Tokyo, Japan) relies on a bandwidth light filter resulting in an increased contrast of the mucosal microsurface and microvessels. Several studies and meta-analyses have demonstrated the efficacy of NBI on the detection, characterization, differentiation, and margin delineation of GC [12,44,45,46,47].

FICE (Flexible spectral imaging color enhancement), developed by Fujifilm (Fujifilm Co., Kanagawa, Japan), consists of a post-selection of a wide range of spectral combinations, improving the resolution and enhancement (Figure 1C). Afterward, Fujifilm developed blue laser imaging (BLI), which uses two different monochromatic lasers to produce a narrow blue band to intensify changes in the mucosal surface. Similarly to NBI, BLI also showed high diagnostic accuracy for the detection and characterization of GC [44,48]. The linked color imaging (LCI), also by Fujifilm, emphasizes the contrast of hemoglobin by expanding the color redness and, therefore, producing bright images that can improve the visibility of lesions [49,50]. In a multicenter randomized trial, the percentage of patients with one or more neoplastic lesions diagnosed with LCI was higher than with WLE (8.0%; 95% CI, 6.2–10.2% vs. 4.8%; 95% CI, 3.4–6.6%), and the proportion of patients with overlooked neoplasms was lower in the LCI group than in the WLE group (0.67%; 95% CI, 0.2–1.6% vs. 3.5%; 95% CI, 2.3–5.0%). The I-SCAN method, developed by PENTAX (Tokyo, Japan), consists of various types of post-processing images to enhance the surface, contrast, and tone. In a comparison trial, i-SCAN showed similar results compared with dye CE (acetic acid and indigo carmine) to delineate the margins of gastric lesions [51].

## 6. Magnifying Endoscopy

Image-enhanced magnifying endoscopy is a tool to better characterize the detected lesion and to correlate it with pathology, helping the endoscopist to differentiate benign from malignant or pre-neoplastic conditions. In a meta-analysis study including 1724 patients and 2153 lesions, the pooled sensitivity, specificity, and area under the curve for the diagnosis of EGC using WLE were 48%, 67%, and 62%, respectively. The use of magnifying endoscopy with NBI improved these rates to 83%, 96%, and 96%, respectively [12].

To diagnose subtle mucosal changes, it is necessary to recognize the normal magnifying features of gastric mucosa. Histologically, there are two different types of gastric glandular epithelium: fundic and pyloric. Normal fundic mucosa is present in the gastric body and fundus of patients without pathological changes, such as inflammation or AG secondary to *H. pylori* infection. Chromoendoscopy with the magnification of normal fundic mucosa is characterized by an epithelial surface where the crypt opening is seen as oval or round surrounded by a white-colored structure (marginal crypt epithelium). Collecting venules have a regular arrangement (RAC) and a greenish (cyan) color. In normal pyloric mucosa, the glands open obliquely and not perpendicularly as in the fundic pattern, resulting in a reticular aspect with grooves. The capillaries form spiral or spring loops (Figure 4):

As stated above, IM is considered a risk factor for the development of intestinal GC. Even though the definitive diagnosis of IM relies on histopathologic evaluation, magnifying endoscopy has also been shown to be an important tool for diagnosis, characterization, and guiding target biopsies. A fine blue-white line on the epithelial surface called the “light blue crest” was described by Uedo et al. and is a good predictor of IM [52].

Yao et al. first described the vessel plus surface (VS) classification, and later, Muto et al. reported the magnifying endoscopy simple diagnostic algorithm for early gastric cancer (MESDA-G) for the differentiation between neoplastic and non-neoplastic lesions [53,54]. The MESDA-G classification relies on the concept that neoplastic changes in gastric mucosa are followed by a clear demarcation line between the normal and the altered tissue. This line is defined as an abrupt change in the surface pattern, either in vascular or glandular structures, as shown in Figure 5.

According to the vs. classification, microvascular and microsurface patterns are classified into three categories: regular (normal), irregular, and absent. The irregular vascular pattern shows a variety of different capillaries or shapes irregularly distributed. When no microvessels are seen, the V pattern is considered absent. This occurs especially when a white opaque substance (WOS) is present in the superficial mucosal layer and prevents the visibility of the epithelium vasculature [55]. In such cases, instead of evaluating the vascular pattern, the distribution type of the WOS is classified as homogeneous or heterogeneous to make a differential diagnosis of cancer and adenoma (Figure 6). The surface pattern (S) is characterized by mucosal glandular structures. When asymmetrically distributed or in various morphologies, it is classified as irregular. If no glandular structure is seen, it is classified as absent. In this case, the V pattern is used for differentiation between neoplastic and non-neoplastic lesions (Figure 7).

The vs. classification showed high accuracy (97%), a positive predictive value (79%), and a negative predictive value (99%) for the diagnosis of intestinal-type EGC [56]. Nevertheless, diffuse- or undifferentiated-type carcinoma characterization may be a limitation for endoscopic diagnosis. This occurs because neoplastic involvement occurs horizontally in the deeper layers of the mucosa without glandular formation, and during the early phase there may be no visible endoscopic alterations, and even biopsies may be negative because the cancer cells are usually deeper and more widespread. This difficulty in detection contributes to a late diagnosis and, consequently, a worse prognosis. In addition, it is also difficult to determine the depth of invasion and distinguish the lateral limits of undifferentiated lesions.

Under WLE, the typical presentation of undifferentiated tumors is a flat or depressed lesion with a pale color. According to Yao, this is due to the reduction in hemoglobin levels [57]. Magnifying endoscopy is important for the evaluation of undifferentiated GC. Usually, in this type, there is no glandular formation, and, consequently, the microsurface pattern is classified as absent. In this scenario, only microvessels can be seen [58] and typically present as a corkscrew pattern with a loop-opened look (Figure 8).

## 7. Artificial Intelligence

Artificial intelligence (AI) has been used to support physicians interpreting medical images. The related applied terminology is computer-aided detection (CADe) and computer-aided diagnosis (CADx), when algorithms are developed to detect and differentiate pathologies, respectively [59]. For this purpose, the most used system is deep learning due to the capacity to be trained with a large number of images and to extract specific clinical features to further predict or classify new images [60,61].

As already discussed, physicians’ ability to adequately diagnose upper GI lesions varies greatly, with a missing rate of EGC of 9.4% (95% CI 5.7–13.1%) [9]. In contrast to colorectal lesions, GC is often subtle (e.g., flat or slightly depressed) and hard to recognize due to the inflammation environment of chronic gastritis [59].

After the publication of the first deep learning-based AI system, which showed promising results with a sensitivity of 92.2% for detecting GC [62], other studies have been conducted, and the commercialization of the first software may not take long. A tandem randomized prospective trial showed a significantly lower miss rate of EGC in the AI-first group than in the routine-first group (6.1%, 95% CI 1.6–17.9% vs. 27.3%, 95% CI 15.5–43%) [63]. Additionally, it was shown that AI was able to predict the superficial cancer invasion depth with a specificity of 78–95% [64]. Consequently, it will help a more homogeneous diagnosis across physicians, improving the decision of referring patients to endoscopic or surgical resection.

In a meta-analysis, the overall sensitivity, specificity, and area under the curve of AI were 89% (95% CI 85–93%), 93% (95% CI 88–97%), and 94% (95% CI 0.91–0.98%), respectively, being comparable to experts and superior to non-experts [64]. Moreover, it is worth mentioning that the use of AI will also play an important role to help in the diagnosis of blind spots and photo documentation with an accuracy of 90.4% [65,66], supporting the quality control of exams and guidelines compliance [32,33,34].

Although other robust prospective trials should be conducted to deeply evaluate the real performance of AI during routine EGD exams, randomized controlled trials evaluating colon AI have proven that this technology is safe to be adopted and potentially improves patient outcomes while reducing costs [67,68,69].

## 8. Other Methods

### 8.1. Confocal Laser Endomicroscopy

Confocal laser endomicroscopy (CLE) is an endoscopic imaging tool that provides cross-sectional real-time in vivo images at the cellular and microvascular levels at a 1000-fold magnification. Fluorescent contrast is required to obtain the imaging, and the most utilized contrast agents are intravenous fluorescein and topically applied acriflavine. Two types of CLE systems have been studied for clinical use: electronic CLE, which integrates a miniature confocal scanner into the tip of a special flexible endoscope, and probe-based CLE (pCLE), a flexible microprobe that can pass through the working channel of a standard endoscope. However, only pCLE is commercially available [70].

CLE allows histopathological diagnoses when utilized by trained endoscopists. Therefore, targeting the region with higher accuracy for a pathology assessment may have a relevant impact on treatment management, avoiding further diagnostic procedures for tissue sampling [71]. The use of pCLE optical diagnosis has been deeply studied for gastric IM, AG, and intraepithelial neoplasia [72,73,74], and it was shown that it contributes to distinguishing normal and neoplastic mucosa through the analysis of both cellular and vascular patterns [75]. In one study, pCLE was able to diagnose AG and IM with 98% specificity [72]. In a recent meta-analysis that included seven studies, pCLE had a sensitivity of 87.9% (95% CI 81.4–92.4%), a specificity of 96.5% (95% CI 91.5–98.6%), and an accuracy of 94.7% (95% CI 89.5–97.4%) to diagnose GC [73].

Despite these encouraging findings, pCLE is not regarded as a standard approach for diagnosis, as adoption is likely reduced for cost reasons and the lack of cost-effectiveness studies. There is also a need to define a framework for how to introduce this tool in the workflow between endoscopists and pathologists [76].

### 8.2. Endocytoscopy

Similarly to CLE, endocytoscopy has the capability of capturing ultra-magnified pathology images when placing the endoscope on the target mucosa during inspection [77]. It provides information about the shape of cells and individual cell nuclei of the mucosa’s superficial layer. For instance, goblet cells for the evaluation of IM and the presence of structural or nuclear atypia to diagnose cancerous lesions, with a specificity of 93.3–100% [77,78].

Endocytoscopy leads to the benefits of the endoscopic optical biopsy but also requires additional training and expertise. Moreover, the need for double staining with crystal violet and methylene blue for optimal evaluation may limit the clinical application because this additional step often increases the duration of the exam, and the staining may not be easily available. However, the use of endocytoscopy with NBI (with no staining) using the vs. classification has been studied with a reported accuracy higher than magnifying NBI [53]. Additionally, it has been shown that this evaluation can be simplified by assessing the microvascular pattern alone, which showed similar accuracy compared to the evaluation of microvascular and microsurface patterns [79]. Recently, Noda et al. [80] studied the application of AI to support the use of endocytoscopy. AI showed a specificity of 90.9% in diagnosing EGC, which was comparable with experts and superior to non-experts.

In summary, endoscopy plays a major role in the diagnosis of gastric cancer. It is important for the endoscopist to be aware of patients at risk, to recognize the endoscopic aspects of early neoplasms, and to understand the advantages and limitations of each endoscopic technique, as well as how to overcome them. Table 1 summarizes the advantages and limitations of the endoscopic techniques for the detection and characterization of gastric cancer.

## 9. Conclusions

Despite the significant improvement in endoscopic imaging for GC over the years, the adequate diagnosis and characterization of EGC is still a challenge. Consequently, GC awareness is crucial and demands a minimum proficiency among physicians, who should be encouraged to use chromoendoscopy. While novel imaging modalities are increasingly being studied, especially for optical diagnosis, specific experience, integration to current workflow, and cost-effectiveness studies should take place for major endorsement.

As modern imaging technology continues to grow, more complexity and time are added to endoscopic clinical practice. To overcome part of these challenges, AI systems will potentially play an important role in supporting physicians with more assertive diagnoses.

## Figures and Tables

**Figure 1 cancers-15-02445-f001:**
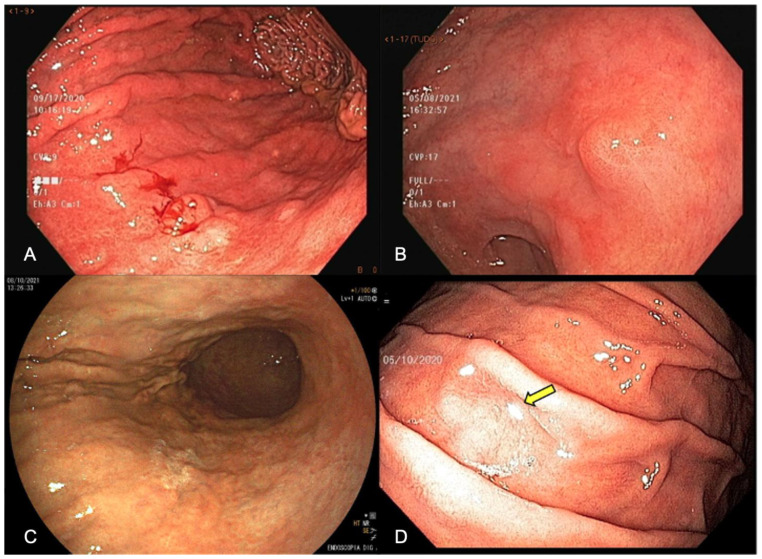
Endoscopic appearance of early gastric cancer (**A**) white-light endoscopy (WLE) showing spontaneous bleeding, (**B**) WLE showing irregular surface, showing a slightly elevated lesion with a central depression, (**C**) pallor color change in the posterior wall of the stomach enhanced by FICE (Flexible spectral imaging color enhancement), (**D**) change in light reflection (arrow) under WLE.

**Figure 2 cancers-15-02445-f002:**
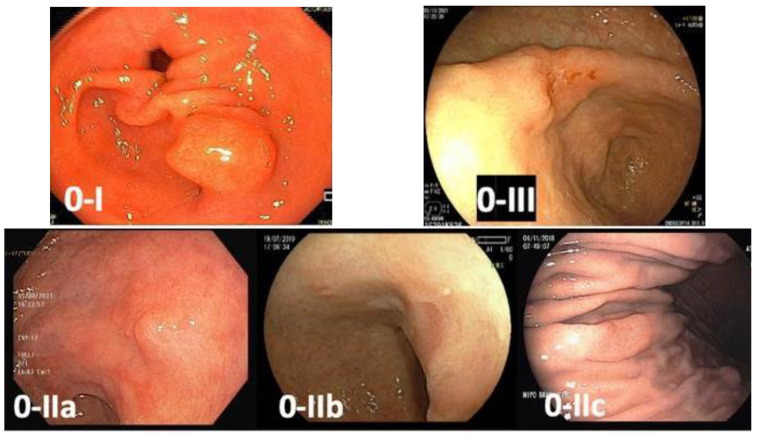
Paris classification of gastric superficial lesions—0-I: polypoid lesion; 0-IIa: flat and slightly elevated lesion; 0-IIb: completely flat lesion; 0-IIc: flat and slightly depressed lesion; 0-III: excavated lesion.

**Figure 3 cancers-15-02445-f003:**
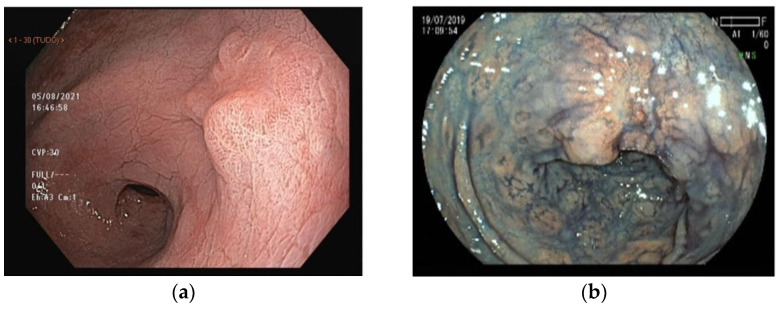
(**a**) White-light endoscopy with acetic acid chromoendoscopy showing an elevated lesion with central depression. The lesion is slightly reddish compared to the surrounding mucosa, suggesting a neoplastic lesion. Biopsy revealed tubular adenocarcinoma; (**b**) chromoendoscopy with indigo carmine dye delineating gastric lesion’s edges (Paris 0-IIa + IIc).

**Figure 4 cancers-15-02445-f004:**
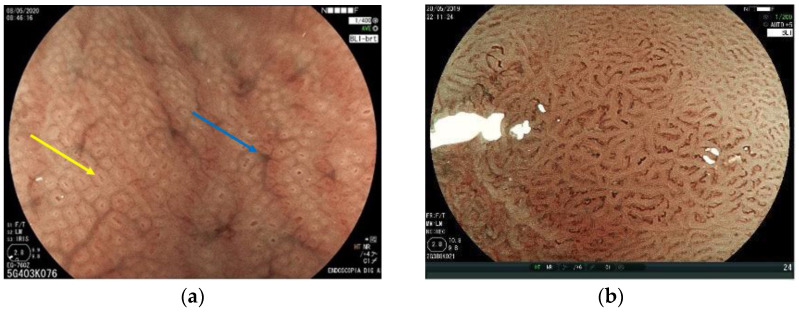
(**a**) High-definition magnifying view of normal fundic mucosa. Note the round pits surrounded by a white-colored structure (marginal crypt epithelium—yellow arrow). The regular arrangement of collecting venules is easily seen (blue arrow); (**b**) normal pyloric gland mucosa shows a reticular aspect with grooves.

**Figure 5 cancers-15-02445-f005:**
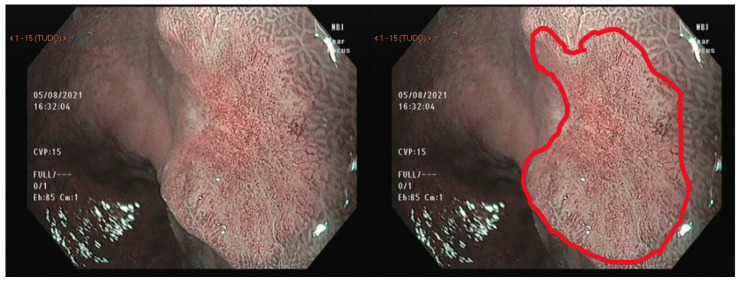
NBI with near-focus magnification showing the demarcation line of a gastric cancer. NBI = narrow-band imaging.

**Figure 6 cancers-15-02445-f006:**
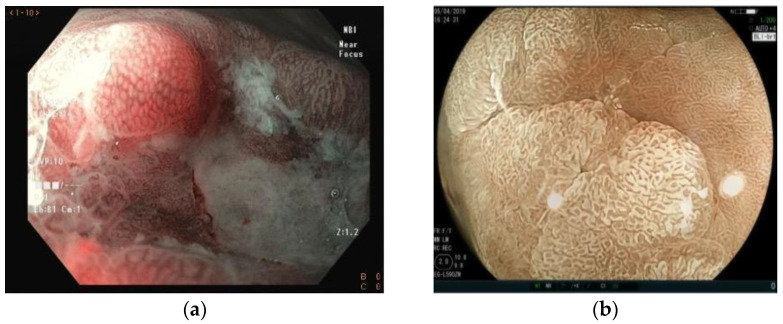
(**a**) Irregular vascular pattern seen with NBI and near-focus magnification; (**b**) absent vascular pattern due to the presence of a white opaque substance inside the demarcation line (BLI—bright view plus magnification). It has a homogeneous distribution, which leads to classifying this lesion as non-cancerous (adenoma). NBI = narrow-band imaging; BLI = blue laser imaging.

**Figure 7 cancers-15-02445-f007:**
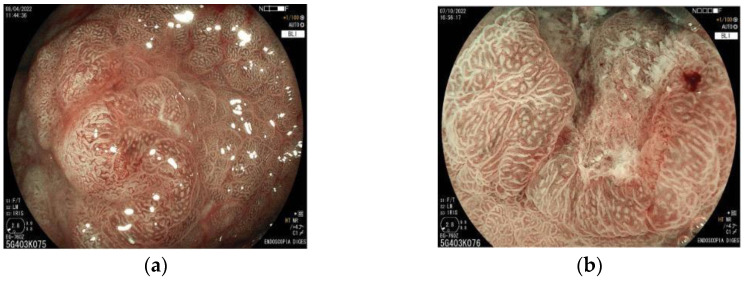
(**a**) Irregular surface pattern and irregular vascular pattern; (**b**) absent surface pattern with a clear demarcation line. BLI with magnification view. BLI = blue laser imaging.

**Figure 8 cancers-15-02445-f008:**
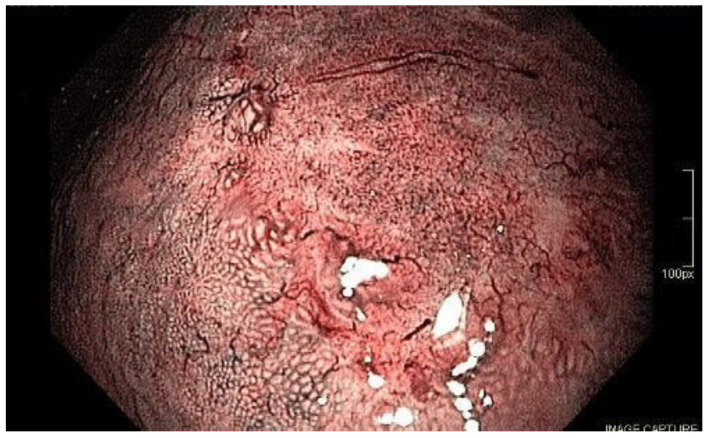
Corkscrew microvessel pattern present on an undifferentiated-type early gastric cancer. Magnified view with NBI.

**Table 1 cancers-15-02445-t001:** Comparison between endoscopic imaging methods for the diagnosis of gastric cancer.

Technique	Advantages	Limitations
White-light endoscopy	Easy to perform Readily available	Low sensitivity and specificity
Dye-based chromoendoscopy	Low cost Widely available Useful for delineating lesions margins	Time-consuming
Virtual chromoendoscopy	Easy to perform Valuable tool for evaluation of microvessels	Low accuracy to predict tumor depth
Magnifying endoscopy	High accuracy to distinguish benign and malignant lesions More specific than WLE and dyes	Low accuracy to distinguish differentiated- from undifferentiated-type adenocarcinoma Limited field of view Low accuracy to predict tumor depth
Artificial intelligence	Real-time diagnosis	High cost Lack of validation from prospective studies
Confocal laser endomicroscopy	Real-time diagnosis	High cost Steep learning curve Limited field of view Need for intravenous or topical contrast
Endocytoscopy	Real-time diagnosis Technology integrated (dedicated endoscope)	High cost Steep learning curve Limited field of view

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
