# Peer review of "Endoscopic Imaging for the Diagnosis of Neoplastic and Pre-Neoplastic Conditions of the Stomach"

_cancers, 2023, doi:10.3390/cancers15092445_

Round 1

Reviewer 1 Report

The authors reviewed the role of endoscopic imaging for the diagnosis of neoplastic and preneoplastic conditions of the stomach. The review is well organized and the bibliography is updated. I have only a few comments.

The authors should reduce the number of abbreviations as it is difficult to follow the main text.

In the introduction it should be worth to mention the role of other imaging techniques in the diagnosis of gastric cancer.

Which are the limitations of each technique? Could the authors compare the different techniques that they are presenting, discussing their advantages and limitations?

Also, it would be useful for the reader if the authors may provide some tables to schematize the main concepts.

Author Response

We would like to thank the reviewers for the attention review and the thoughtful comments which helped us to improve our manuscript.

Reviewer 1

The authors reviewed the role of endoscopic imaging for the diagnosis of neoplastic and preneoplastic conditions of the stomach. The review is well organized and the bibliography is updated. I have only a few comments.

Answer: We thank the reviewer for this comment.

The authors should reduce the number of abbreviations as it is difficult to follow the main text.

Answer: Thanks for this suggestion. We have reduced the number of abbreviations throughout the text.

In the introduction it should be worth to mention the role of other imaging techniques in the diagnosis of gastric cancer. Which are the limitations of each technique? Could the authors compare the different techniques that they are presenting, discussing their advantages and limitations? Also, it would be useful for the reader if the authors may provide some tables to schematize the main concepts.

Answer: We thank the reviewer for this comment. We added a table comparing the advantages and limitations of the different imaging techniques discussed in the article.

Reviewer 2 Report

The manuscript provides a thorough review of the endoscopic imaging used for diagnosis of gastric neoplasia and pre-neoplasia. Communicated knowledge is important and it provides a valuable contribution to the field. In addition, the manuscript is good fit for the special issue on “The Application of Endoscopy in Gastrointestinal Cancers.”

Overall, the manuscript reviewed published data about different gastric endoscopic imaging approaches. However, I think there is a lot of redundancy, and the manuscript lacks a proper flow. The review can be improved if the authors organize the manuscript better and summarize literature in a more concise manner. One sentence paragraphs should be avoided. In addition, I think that providing a table that summarizes pros and cons of each technique would be beneficial.

The manuscript also needs to improve English language and focus on improving sentence structure and spelling. Please make sure that you provide an explanation for each abbreviation and what the numbers indicate. For example (not limited to only this example):

In a meta-analysis study, including 1724 patients and 2153 lesions, the pooled sensitivity, specificity, and AUC for the diagnosis of EGC using WLE were 0.48, 0.67 and 0.62, respectively.” What is AUC and what these numbers should represent?

The legends of the figures should also include more information to better describe them. In particular, it would be great to further describe Figure 1 – Does the green component represent H. Pylori, and if so, how does it play into gastric carcinogenesis, and into the Pelayo correa’s model of carcinogenesis? I think that the image needs improvement in the quality of the image.

Please check the confidence intervals and the statistical analysis numbers when comparing the two techniques. For example (not limited to only these examples):

“In a metanalysis, EGD screening detection of GC (0.55%, 95% CI 0.39 − 0.75%) and early-GC (EGC) (0.48%, 95% CI 0.34 − 0.65%) was superior than radiography screening (GC 0.19%, 95% CI 0.10 − 0.31%; EGC 0.08%, 95% CI 0.04 − 0.13%).” Are these indeed percentages? Explain what the numbers indicate.

“A tandem randomized prospective trial showed a significantly lower miss rate of EGC in AI-first group than in the routine-first group (6.1%, 95% CI 1.6–17.9% vs 27.3%, 15.5–43%)” What is the confidence interval for the second portion?

Author Response

We would like to thank the reviewers for the attention review and the thoughtful comments which helped us to improve our manuscript.

Reviewer 2

Comments and Suggestions for Authors

The manuscript provides a thorough review of the endoscopic imaging used for diagnosis of gastric neoplasia and pre-neoplasia. Communicated knowledge is important and it provides a valuable contribution to the field. In addition, the manuscript is good fit for the special issue on “The Application of Endoscopy in Gastrointestinal Cancers.”

Answer: We thank the reviewer for this comment.

Overall, the manuscript reviewed published data about different gastric endoscopic imaging approaches. However, I think there is a lot of redundancy, and the manuscript lacks a proper flow. The review can be improved if the authors organize the manuscript better and summarize literature in a more concise manner. One sentence paragraphs should be avoided. In addition, I think that providing a table that summarizes pros and cons of each technique would be beneficial.

Answer: We thank the reviewer for this comment. We rearrange some paragraphs as suggested and we added a table summarizing the advantages and limitations of the different imaging techniques discussed in the article.

The manuscript also needs to improve English language and focus on improving sentence structure and spelling.

Answer: We made a throughout review in the manuscript focusing in improving English grammar and structure.

Please make sure that you provide an explanation for each abbreviation and what the numbers indicate. For example (not limited to only this example):

In a meta-analysis study, including 1724 patients and 2153 lesions, the pooled sensitivity, specificity, and AUC for the diagnosis of EGC using WLE were 0.48, 0.67 and 0.62, respectively.” What is AUC and what these numbers should represent?

 Answer: We thank the reviewer for this comment. As suggested by the other reviewer, we reduced the number of abbreviations and area under the curve is non-abbreviated now. Many meta-analysis express pooled results in the interval of 0-1.0 as in this case. We preferred to maintain the original authors pattern in our manuscript, but as suggested by the reviewer, we modified the numbers by percentages to make it more clear to the reader. Thanks for the suggestion.

The legends of the figures should also include more information to better describe them. In particular, it would be great to further describe Figure 1 – Does the green component represent H. Pylori, and if so, how does it play into gastric carcinogenesis, and into the Pelayo correa’s model of carcinogenesis? I think that the image needs improvement in the quality of the image.

Answer: We agree with the reviewer. Figure 1 was suppressed from the manuscript as it didn’t add significant information to the main subject of the article. All figure legends were reviewed and completed with more information.  We thanks for this comment.

Please check the confidence intervals and the statistical analysis numbers when comparing the two techniques. For example (not limited to only these examples):

“In a metanalysis, EGD screening detection of GC (0.55%, 95% CI 0.39 − 0.75%) and early-GC (EGC) (0.48%, 95% CI 0.34 − 0.65%) was superior than radiography screening (GC 0.19%, 95% CI 0.10 − 0.31%; EGC 0.08%, 95% CI 0.04 − 0.13%).” Are these indeed percentages? Explain what the numbers indicate.

“A tandem randomized prospective trial showed a significantly lower miss rate of EGC in AI-first group than in the routine-first group (6.1%, 95% CI 1.6–17.9% vs 27.3%, 15.5–43%)” What is the confidence interval for the second portion?

Answer: We reviewed all confidence intervals of the article. In the first example they are indeed representing percentages. 95% CI was clarified in the second example. All CI throughout the text were reviewed and adjusted when necessary.

Round 2

Reviewer 2 Report

I thank the authors for addressing the comments of the first round of revisions. They have done an extensive work to improve the manuscript. I appreciate adding a summary table of comparing the advantages and limitations for each technique. 

I do, however, have one minor comment about the table itself. It seems to me that the table has two legends, one below and one above it. I suggest that one is removed.  

Author Response

Thank you for the comment. Actually, the first sentence was a paragraph to introduce the table. We expanded this paragraph summarizing the discussion and citing the table. We think it became better and more clear to the reader this way. Thank you.